# CO_2_ Stunning in Pigs: Physiological Deviations at Onset of Excitatory Behaviour

**DOI:** 10.3390/ani13142387

**Published:** 2023-07-23

**Authors:** Bente Wabakken Hognestad, Nora Digranes, Vigdis Groven Opsund, Arild Espenes, Henning Andreas Haga

**Affiliations:** 1Department of Companion Animal Clinical Sciences, Faculty of Veterinary Medicine, Norwegian University of Life Sciences, 1433 Ås, Norway; nora.digranes@nmbu.no (N.D.); andreas.haga@nmbu.no (H.A.H.); 2Department of Production Animal Clinical Sciences, Faculty of Veterinary Medicine, Norwegian University of Life Sciences, 1433 Ås, Norway; vigops@live.no; 3Department of Preclinical Sciences and Pathology, Faculty of Veterinary Medicine, Norwegian University of Life Sciences, 1433 Ås, Norway; arild.espenes@nmbu.no

**Keywords:** carbon dioxide, CO_2_, stunning, pig, aversive behaviour, slaughter, consciousness

## Abstract

**Simple Summary:**

Stunning by carbon dioxide inhalation is commonly used in pigs before slaughter. However, the method is controversial because seemingly aversive behaviours are observed before the pigs appear to lose consciousness, i.e., fall over. It is unknown whether the aversive behaviours occur while pigs are conscious. The aim of the current study was to characterise behaviours observed during carbon dioxide stunning and relate these to physiological variables recorded when each behaviour began. We recorded physiological variables we considered to be essential in maintaining consciousness, and at a high sampling frequency. In total, 13 behaviours were characterised, 7 of which were examined more closely. The results show that some aversive behaviours were observed while physiological variables were both compatible and incompatible with consciousness. However, the physiological variables recorded at the onset of the behaviours “continuous neck dorsiflexion” and “agonal gasping” were sufficiently deranged in all pigs to justify the conclusion that these animals were unconscious when the behaviours began. These findings are important when considering animal welfare in relation to current stunning methods.

**Abstract:**

Stunning by carbon dioxide (CO_2_) inhalation is controversial because it is associated with vigorous movements and behaviours which may or may not be conscious reactions. Furthermore, it is unknown whether some behaviours might indicate the transition into unconsciousness. Our study objective was to investigate the loss of consciousness during CO_2_ stunning by linking physiological variables (in particular pH, PaO_2_ and PaCO_2_) to the onset of observed behaviours. A total of 11 cross-bred pigs were studied. A tracheostomy tube, venous and arterial cannulae were placed under sevoflurane anaesthesia. After recovery from this, and a “wash out” period of at least 30 min, arterial blood samples were taken (and baseline values established) before 90–95% CO_2_ in medical air was administered through the tracheostomy tube. Subsequent behaviours were video-recorded and key physiological variables were evaluated using an anaesthetic monitor and the frequent sampling of arterial blood (albeit with inconsistent inter-sample intervals). After the study, behaviours were classified in an ethogram. At the onset of behaviours categorised as “vigorous movement extremities”, “opisthotonos” and “agonal gasping” pH values (range) were: 6.74–7.34; 6.66–6.96 and 6.65–6.87, while PaCO_2_ (kPa) was 4.6–42.2, 24.4–51.4 and 29.1–47.6. Based upon these values, we conclude that the pigs were probably unconscious at the onset of “opisthotonos” and “agonal gasping”, but some were probably conscious at the onset of “vigorous movements”.

## 1. Introduction

Stunning by CO_2_ inhalation before slaughter is used extensively in commercial abattoirs. In Norway, the practice involves the immersion of groups of pigs into chambers, usually pre-filled with 89–93% CO_2_, for 135–180 s. Inhalation of CO_2_ causes acute hypercapnia and respiratory acidosis that rapidly reduces intracellular pH throughout the central nervous system [1,2]. The method has several advantages. Pigs can be managed in groups with minimal handling by humans, which reduces stress. The method also entails only a minor risk of operator errors and offers reliable stun quality; there is little risk of animals regaining consciousness if the exposure time is sufficient. Despite this, there are animal welfare concerns with CO_2_ stunning. The basis of these concerns is the pain associated with CO_2_ inhalation and the potentially associated aversive behaviours animals show on exposure. Previous studies have shown that high concentration of CO_2_ is aversive in pigs, induces respiratory distress, and may elicit a fear response [3,4,5,6]. Exposed pigs often vocalize and have vigorous muscle activity before the loss of posture. Importantly, the time from CO_2_ exposure to loss of consciousness is still undetermined. In 2020, the European Food Safety Authority (EFSA) published an independent scientific report on the slaughter of pigs, in which several welfare aspects of CO_2_ stunning were examined [7]. The EFSA concluded that CO_2_ stunning of pigs can cause pain, fear, and respiratory distress and it does not cause an immediate loss of consciousness. Behavioural observations, electroencephalography (EEG), blood gas analyses as well as neurologic reflex testing have been used in an attempt to determine the time to loss of consciousness [8,9,10,11,12,13,14,15,16]. The EEG is an objective measure of brain activity with an isoelectric EEG indicating unconsciousness. However, the exact time of transition from consciousness to unconsciousness cannot be determined using contemporary EEG methods, as there are considerable challenges with movement artefacts associated with vigorous muscular activity [13,14,15,16,17]. Consequently, previous electroencephalographic studies show little agreement in attempts to link consciousness with muscular activity during CO_2_ stunning. 

A state of consciousness requires that certain features of the brain’s milieu intérieur, e.g., its pH, must be maintained within critical limits. Additionally, the circulatory system must supply sufficient oxygen. Such dependency justifies the use of arterial blood sampling in previous studies [2,9,18,19]. However, it is technically challenging not only to sample arterial blood from pigs in the stunning environment but to be able to reliably link samples with rapidly changing behavioural events.

Cardiovascular variables, i.e., mean arterial blood pressure and heart rate have been studied during CO_2_ stunning in pigs [15,20,21]. Ventilatory variables beyond a visual assessment of respiratory pattern and frequency have not been studied. Arterial pH, gas tensions, and mean arterial blood pressure are affected by CO_2_ inhalation, and major derangements may render the brain incapable of maintaining consciousness. Previous studies have statistically analysed these variables in terms of population indices of central tendency, i.e., means and or median values. However, we believe that from a welfare position, a case may be made for individual pigs to be studied. 

On the basis of this, we decided to investigate pig behaviours exhibited during CO_2_ stunning and the physiological variables related to those behaviours in individual animals. The primary aim of the current study was to describe key physiological variables at the onset of behaviours observed in pigs during CO_2_ inhalation. A secondary aim was to provide a more detailed description of the physiological changes induced by CO_2_ stunning by taking more samples at a greater frequency than hitherto has been attempted. 

## 2. Materials and Methods

The experiments were performed between the 24th of November and the 2nd of December 2021 at the research animal facility of the Norwegian University of Life Science. Ethical approval was provided by the Norwegian Food Safety Authority (FOTS ID 28418), Oslo, Norway on 8 November 2021.

### 2.1. Animals and Housing

In total, 11 commercial cross-bred pigs (25% Norwegian landrace, 25% Yorkshire and 50% Duroc), 5 surgically castrated males and 6 intact females with body weights of 24–32 kg on the day of the experiment were studied. Since the aim of the study was to record physiological variables at the onset of behaviours in individual pigs in order to elucidate whether the animals were unconscious or not, we determined the number of pigs to be studied based on the power to identify at least one individual with variables indicating consciousness—if such a subgroup of individuals existed. By using simple probability calculations, the power of the study to find at least one pig showing a behaviour while being conscious was calculated to be 83% if at least 15% of the background population belonged to this subgroup. The pigs had no history of treatment for systemic disease and originated from the Livestock Production Research Centre at the Norwegian University of Life Science. They were from four different litters, but all were 38 days old at the time of arrival. The pigs were housed in pairs in concrete pens with a mixture of wooden shavings and peat (Naturtorv, Floralux) as bedding. Humidity and temperature were controlled and kept at 40–50% and 18–20 °C, respectively. A 12:12-h light/dark cycle was used with a 30-min transition period. The pigs were fed on a commercial pig diet (Format Vekst 110, Felleskjøpet, Norway) and hay. Water and food were available ad libitum, but the pigs were fasted approximately 12 h before beginning the experiment. The pigs were handled daily and trained to enter the transport trolley by using positive reinforcement. Pens were environmentally enriched with a ball and a Kong (The KONG Company EU Ltd., Groß-Gerau, Germany). The pigs were acclimatised for at least 6 days before the experiment started.

### 2.2. Experimental Procedure

This was a prospective descriptive study. On the first experimental day, the pigs were clinically assessed and judged to be healthy. They were weighed and then transported individually from the pen to the surgical area at the Research Animal Facility. Anaesthesia was induced and maintained with sevoflurane in O_2_ with a tight-fitting mask (Anestesimaske Large; Kruuse, Norway). After topical laryngeal application of lidocaine (Xylocaine 100 mg/mL spray, Aspen, Denmark), the pig was endotracheally intubated (Parker Flex-Tip ID 5–6). A circle rebreathing system was attached before the imposition of volume-controlled intermittent positive-pressure ventilation at a rate of 20 breaths per minute using an anaesthetic machine (Dräger Perseus A500, Dräger Norge AS, Oslo, Norway). The tidal volume was regulated to achieve an end-tidal CO_2_ between 4.0–4.5 kPa. A 22G peripheral venous catheter (Venflon Pro; Becton Dickinson Infusion Therapy, USA) was placed in the lateral auricular vein and secured. Arterial cannulae (20 SWG; 16 cms) (Arterial Catheterisation set, Arrow Int. Inc., USA) were placed in both tibial arteries using the Seldinger technique. One of the arterial catheters was connected to a pressure transducer (TruWave pressure monitoring transducer; Edwards Lifesciences Corp., USA) fixed at the level of the thoracic inlet and zeroed to atmospheric pressure. An anaesthetic monitor (GE Carescape Monitor B650; GE Heathcare, Helsinki, Finland) was used to record invasive arterial blood pressure, pulse rate (PR), pulsoximetry (SpO_2_), and electrocardiography (ECG) by using 3 leads attached to the dorsal, left lateral and ventral thorax (Red dot electrodes, 3M). Additionally, we performed side-stream multi-gas analysis, conducted spirometry with a pitot tube (Intersurgical, Paediatric Spirometry sensor) and assessed oesophageal temperature. Data were downloaded every 5 s using data collection software (iCollect Version 5.0, GE Healthcare, Helsinki, Finland). 

Ketoprofen (Comforion vet. 100 mg/mL, Orion Corporation, Espoo, Finland) 3 mg/kg was administered intravenously. After infiltration of 1 mg/kg bupivacaine (Marcain 5 mg/mL, Aspen Pharma Trading Ltd., Ireland) a tracheotomy was performed in the cranioventral midline of the cervical area. The orally placed endotracheal tube was pulled rostrally. Subsequently, a guide wire was passed 5 to 10 cm into the trachea and an armoured endotracheal tube (RüsohFlex, Teleflex medical ID 5.0–6.5 mm) was threaded over the wire, ending at the premeasured position of the thoracic inlet. The cuff was inflated, and the breathing system reconnected. The tracheostomy tube was sutured to the skin and then positioned laterally on the cervical area covered with an adhesive bandage (Allevyn, Smith + Nephew). The skin over the surgical site was closed with two interrupted cruciate sutures. After the successful completion of the tracheotomy, anaesthesia was discontinued, and the pig was positioned in a hammock to recover. The hammock was purpose-built for this study, measuring 110 cm in length and consisting of a stainless-steel frame, with a PVC-reinforced canvas sling, four holes for the legs and one strap above the pig’s back. During recovery, the pig was offered food rewards in an attempt to limit stress.

When a minimum of 30 min had passed and the end-tidal sevoflurane concentration was <0.3%, baseline data were collected over 5 min. A video camera (Panasonic 4K professional, model No. AG-CX350E, Japan) was positioned on the left side of the pig to record a close lateral view, with a second (identical) device placed on the right side to record a panoramic view. Video recordings were started, and arterial blood was anaerobically sampled into a heparinised syringe (Pico 70 Self-fill; Radiometer, Denmark) a maximum of 2 min before CO_2_ administration began. An anaesthetic machine (Dameca 10750, Denmark) was used to deliver a mixture of CO_2_ and air. The N_2_O flowmeter was connected to pressurised CO_2_ and the flow was set to 6 L/min. Medical-grade air was administered via the O_2_ flowmeter starting at 1 L/min. The CO_2_ concentration was measured and regulated before each experiment (CheckMate 3, Dansensor, MOCON Europe A/S, Denmark) to achieve an inspired CO_2_ concentration of 90–95%. This was delivered directly to the tracheostomy tube using a Bain breathing system. Immediately after CO_2_ administration began, attempts to take arterial blood samples at regular intervals were made, although vigorous muscular activity prevented regular sampling. Nevertheless, the precise time was recorded for all samples, which were analysed within 30 min using a benchtop blood gas analyser (ABL 800 Flex; Radiometer, Denmark). When no cardiac electrical activity was recorded or a minimum of 10 min had elapsed after CO_2_ administration had begun, 100 mg/kg pentobarbital (Euthasol 400 mg/mL, Le Vet BV, Oudewater, The Netherland) was administered intravenously. A post-mortem examination was performed on all pigs within one hour of death. The experimental timeline from first anaesthesia until postmortem examination is illustrated in Figure 1. Histopathology was performed on samples of lung, liver, kidney and myocardium from each animal. The post-mortem examination and histopathology evaluation was conducted by a veterinary pathologist with 30 years of experience. The brain was examined in seven of the animals, in chronological order on animals no. 2, 3, 4, 7, 8, 10 and 12. Samples of tissues were fixed for at least three days in formalin, routinely processed by dehydration in ethanol and xylene and embedded in paraffin. Half of the brain was fixed in whole for at least one week before sampling of 3 mm thick pieces from the brainstem, cerebellum, thalamus, hippocampus, caudate nuclei and the cortex of the temporal lobe. Four micrometre-thick sections were stained with haematoxylin and eosin (HE). 

### 2.3. Data Collection and Processing

The video recordings were evaluated at least twice by one of the authors. All the videos included the baseline period and a minimum of five minutes of CO_2_ administration. The observer was aware of each animal’s identification and the timing of CO_2_ administration. On the first viewing, behaviours were described and categorised in an ethogram for objective assessment. The recordings were then re-evaluated and the timing of specific behaviours was recorded for each animal. 

Physiological variables were recorded as in the first anaesthesia. Heart rate was taken principally from the invasive blood pressure and secondarily from the ECG. The ECG was visually evaluated, and the rhythms observed were categorised retrospectively. Spurious numerical data were identified by simultaneous inspection of the video recordings and physiological traces on the anaesthetic monitor. A database was constructed with a 5 s resolution in Excel (Microsoft Excel Version 2209 Build 16.0.15629.20256). For physiologic data with a resolution of >5 s, the values at the onset of each behaviour were estimated by linear interpolation.

## 3. Results

The mean (standard deviation) anaesthesia time for instrumentation was 123 (18) min. Before CO_2_ inhalation began, 9 pigs were calm, and all exhibited the spontaneous opening and closing of eyelids. The two remaining pigs were restless and required more food rewards to remain still. After CO_2_ administration began, the spontaneous blinking rate became impossible to establish because of vigorous movements. The behaviours observed and characterised in the first evaluation are shown in Table 1. Based upon the number of animals exhibiting specific behaviours, the following behaviours were more closely examined: “slow head movement”, “gasping”, “slow extremity movement”, “vigorous extremity movement”, “vigorous head movement”, “opisthotonos” and “agonal gasping”. The number of pigs exhibiting these behaviours at various time points from the start of CO_2_ inhalation is illustrated in Figure 2.

Table 2 gives the time to the onset of the investigated behaviours and the corresponding values for selected physiological variables. Figure 3 illustrates PaCO_2_ and pH for each individual pig at onset of the investigated behaviours. Vigorous extremity movement was observed early in pigs 5 and 12, within 12 and 5 s, respectively, with a PaCO_2_ value of 13.54 kPa and 4.61 kPa and a pH value of 7.107 and 7.338. Vigorous head movement was observed early in pig 12 at 5 s, with a PaCO_2_ value of 4.61 kPa and pH 7.338. 

The recorded physiologic data are given in Figure 4 and Figure 5. Due to vigorous muscular movements, the waveforms of ECG and arterial blood pressure were obscured at certain time points, rendering data collection incomplete.

Second-degree atrioventricular block was observed during baseline recordings in one pig. This was also seen in four other pigs during CO_2_ inhalation. Ventricular premature complexes were observed during baseline in one pig and in two other pigs upon inhaling CO_2_. Ventricular escape beats were observed in five pigs inhaling CO_2_. Pulseless electrical activity was observed in all 11 pigs when inhaling CO_2_. During stunning, red, frothy mucus was observed in the tracheostomy tube of pigs number 3 and 12.

In pig 2 the behaviours “slow head movement”, “slow extremity movement”, “vigorous extremity movement” “opisthotonos” and “agonal gasping” outlasted these in the other animals by 100, 80, 148, 74 and 80 s, respectively. In pig 2, the fraction of inspired oxygen was 0.10, indicating an undetected leak was allowing partially inspiration of room air. This pig is excluded from Table 2 but is included in Figure 2, Figure 3, Figure 4 and Figure 5.

Due to vigorous movement in pig 3, the breathing system became disconnected 30 s after t = 0 but was reconnected within 10 s. Pig 3 did not define the range for the time to start of any of the behaviours given in Table 2. 

The post-mortem examination revealed that all pigs had pulmonary congestion with oedema. All the pigs also showed various degrees of haemorrhage in the alveoli, pulmonary interstitium and conducting airways. Pigs number 3 and 11 had interstitial pneumonia that was probably present before the experiment.

Of the 11 pigs studied, 7 were randomly selected for histological preparations of the CNS. In total, 4 of these pigs exhibited mild, acute haemorrhage in the brainstem. The other 3 pigs did not reveal macroscopic or microscopic changes in the examined sections of the CNS. 

## 4. Discussion

Figure 2 indicates that when CO_2_ inhalation began, the overall order of behaviours was that “gasping”, “slow extremity movement” and “vigorous extremity movement” occurred before “opisthotonos” and “agonal gasping”. In some instances, “gasping” and “slow extremity movement” occurred almost immediately after the CO_2_ administration began. “Vigorous head movement” occurred early in one pig, but generally it began after the onset of “gasping” and “slow extremity movement”. The behaviours “opisthotonos” and “agonal gasping” occurred first at 20 s and 31 s after CO_2_ inhalation began (Table 2), which may indicate that these behaviours mark the onset of another phase of the stunning process.

The pH and PaCO_2_ values in several pigs at the onset of “gasping” and “slow extremity movement” behaviours were considered to be compatible with consciousness. Figure 3 reveals the pH values were 6.8–7.3 and 6.8–7.5 and PaCO_2_ was 5.5–42.1 kPa and 4.4–33.6 kPa, respectively, for the two behaviours. For “vigorous extremity movement” and “vigorous head-movement” two pigs, numbers 5 and 12, had pH and PaCO_2_ values considered to be compatible with consciousness. If the results of Eisele et al. [22], are accepted (these show the progressively anaesthetising effect of CO_2_ when PaCO_2_ reaches 13 kPa) then pig 5 was probably in transition between consciousness and unconsciousness when “vigorous extremity movement” and “vigorous head movement” began. Therefore, vigorous behaviour may not be a suitable indicator that unconsciousness is present. At the onset of the behaviours “opisthotonos” and “agonal gasping”, the physiological changes were so deranged that no pigs were likely to be conscious at this time.

The terms “slow extremity movement”, “vigorous extremity movement” and “vigorous head movement” were used to denote forms of activity that have been previously studied during CO_2_ inhalation in pigs [16,18,23,24]. “Vigorous extremity movement” in the current study is probably similar to the “muscular excitation” referred to by Atkinson et al. and Rodriguez et al. [18,23]. If that is so, their results are in agreement with those of the current study regarding the timing of the appearance of vigorous behaviours in relation to CO_2_ stunning. However, pig 5 and pig 12 in the current study deviated from this pattern and showed an early onset of vigorous behaviour.

“Gasping” behaviours are commonly described in behavioural studies of CO_2_ immersion and are observed before muscular excitation [16,18,24]. They were also seen in the current study. Previous studies also describe “gagging” which occurs later in the stunning process [18,23,24]. However, such reports do not indicate whether ventilation in between “gagging” has ceased or not which makes it difficult to determine if it is similar to the “agonal gasping” observed in the current study. If respiratory patterns such as agonal gasping are to be used to assess consciousness it is important to distinguish between spontaneous open-mouth breathing, as occurs early in the stunning process, and true agonal gasping. 

“Opisthotonos” and “agonal gasping” occurred after “gasping”, “slow extremity movement”, “vigorous extremity movement” and “vigorous head movement” in the current study. Both behaviours are commonly observed in moribund subjects considered to be unconscious [25]. In the current study the onset of “opisthotonos” and “agonal gasping” were linked with PaCO_2_ and arterial pH values that supported the likelihood that all pigs were unconscious [22].

The onset of CO_2_ inhalation was associated with a rapid increase in arterial PaCO_2_ and a rapid decrease in arterial pH. By 20 s, PaCO_2_ values exceeded 20 kPa and arterial pH values were below 7.0 in all pigs. This broadly agrees with the results of previous studies [2,9,18] even though the method of CO_2_ delivery differed. The rapid change in pH and PaCO_2_ might indicate that the pigs in the current study ventilated at a level more equivalent to the real situation. In humans, most individuals will be unconscious when the end-tidal concentration of an inspired volatile anaesthetic is 50% of the minimum alveolar concentration (MAC) of that agent [26]. The dose differential required for inducing unconsciousness as opposed to motor responses may be even greater in pigs. Using EEG, partial cerebrocortical electrical silence has been observed at isoflurane concentrations < 1 MAC in pigs [27]. This indicates that in pigs anaesthetised with isoflurane, a lower arterial concentration is needed to attain cerebrocortical depression than is required to inhibit movement caused by noxious input. However, the mechanisms whereby CO_2_ and isoflurane produce unconsciousness are not the same, therefore it cannot be assumed that CO_2_ produces unconsciousness at 0.5 MAC. A previous study in dogs found that PaCO_2_ of 22 kPa and 32 kPa produces a 50% and 100% reduction, respectively, for halothane [22]. Hence, the PaCO_2_ level at 20 s in the current study may represent a point at which CO_2_ is beginning to act as an anaesthetic. Previous studies have demonstrated that intracellular pH in the central nervous system falls in parallel with arterial blood levels during CO_2_ inhalation [2,22]. A repetition of Eisele et al. study using pigs is required to elucidate the potency of CO_2_ as an anaesthetic in this species.

Figure 4 shows that PaO_2_ decreased during CO_2_ inhalation. In pig 2 PaO_2_ changed less than in other pigs because a suspected leak in the breathing system allowed air dilution. Despite the fact that PaCO_2_ exceeded 25 kPa, pig 2 maintained motor activity for a longer period than other pigs, indicating that low PaO_2_ contributes to the loss of motor activity.

The mean arterial blood pressure (MAP) stayed > 60 mmHg for at least 28 s after CO_2_ inhalation began in all pigs (Figure 5). This is in accordance with both Mullenax et al. [21] and Stengl et al. [28] who recorded initial increases in both arterial and venous blood pressure after CO_2_ inhalation began in pigs. Cerebral perfusion is dependent upon MAP and either jugular venous or intracranial pressure (ICP) depending on which is greater. In the current study, neither jugular venous pressure nor ICP were measured. However, when MAP values exceed 60 mmHg, the ICP or jugular venous pressures must be severely elevated to compromise cerebral perfusion. A MAP value > 60 mmHg is therefore likely to indicate that cerebral perfusion pressure was adequate for the first 28 s at least, in all pigs. Therefore, it seems unlikely that circulatory failure was the cause of unconsciousness in these pigs.

Martoft et al. found that heart rates did not change noticeably during the first 10 s of CO_2_ stunning, but then declined progressively and decreased to 10 ± 10 beats minute^−1^ at 60 s when 90% CO_2_ was breathed. Indeed, in 9 of 12 trials, the heart had stopped beating by that time [2]. These findings contrast with the work of Terlouw et al. who found a gradually increasing heart rate after exposure to 80% CO_2_ [20]. Stengl et al. also reported a significantly increased heart rate in experimentally induced hypercapnic acidosis [28]. Mullenax et al. observed an initial decrease in heart rate from baseline, which they attributed to a pre-treatment heart rate elevated by restraint. There followed a gradual increase in heart rate for 90–120 s, before a marked decline [21]. In the current study, heart rate did not show a marked decline by 60 s and only one pig exhibited asystole at 60 s. Hypercapnic acidosis is known to increase both norepinephrine and epinephrine in pigs [29], and probably contributed to maintaining heart rate within normal levels for the first 60 s during CO_2_ inhalation.

Respiratory minute volume increased in all pigs after CO_2_ inhalation began, except in pig 9. These results are consistent with previous findings [9] and illustrate the physiological response to hypercapnia. Eisele et al. also demonstrated an initially increased respiratory minute volume in CO_2_ anaesthetised dogs [22]. At 60 s minutes volume decreased in most pigs in the current study (except for pig 2) and probably represents a point where cerebral acidosis compromises respiratory centre function. Pig 9 had a high respiratory minute volume before CO_2_ inhalation began.

To our knowledge, there have been no published post-mortem reports on CO_2_ stunning in pigs beyond the assessment of meat quality [30]. The main reason for doing so in the current study was to detect subclinical diseases that could have affected results. In the current study, post-mortem examination revealed all pigs developed pulmonary congestion with oedema. This was also reported in humans after CO_2_ intoxication [31]. Furthermore, various degrees of haemorrhage were also observed in the alveoli, pulmonary interstitium and conducting airways. The cause for this may have been related to hypoxic pulmonary vasoconstriction. Two pigs with pre-existing pneumonia did not differ from the others in terms of physiological variables, which is in agreement with a previous study [11]. In four pigs, mild brainstem haemorrhage was observed although the cause is unknown. A detailed analysis of post-mortem findings is beyond the scope of this article.

The focus of the current study was to describe the response of individual pigs to CO_2_ inhalation, rather than find indices of central tendency for populations. This is because individual animals experience CO_2_ stunning. In the current experiment, even had the measure of central tendency for any given physiologic variable changed significantly at the onset of specific behaviour, this would not necessarily have meant that all animals had reached similar levels. If one were to ensure that not even a small proportion of the population showed the behaviour while corresponding physiologic variables remained within normal limits, then a considerably greater number of animals would need to have been studied.

During the experiment, we did not experience any major problems with the model and successfully collected the data as intended. However, there were challenges with the breathing system which became transiently disconnected in one pig because of excessive movement.

In the current study, a tracheostomy was performed in order to accurately deliver the desired CO_2_ concentration and identify the onset of inhalation precisely. Consequently, CO_2_ bypassed the upper respiratory tract meaning that aversive behaviours caused by pain in the upper respiratory mucosae (which is what may be seen in the real-life situation) were not observed. Therefore, we propose that the aversive behaviours observed from the onset of CO_2_ inhalation to the loss of consciousness are related to “air hunger” caused by increasing PaCO_2_ rather than pain due to mucosal irritation.

In contrast to the real-life situation, pigs in the current study were immediately exposed to at least 90% CO_2_ making it difficult to match study events to similar time points in the stunning process (in which pigs spontaneously breathe CO_2_ from the chamber). However, the order of behaviours exhibited will probably be similar—especially for reflex behaviours. This subject might be further researched by video-recording behaviour during actual stunning. 

The time to the onset of opisthotonos and agonal gasping, which corresponded with low pH and high PaCO_2_ values (and which were probably incompatible with consciousness), could in the future be recorded in a larger number of pigs. This would allow the distribution of times to effect of CO_2_ inhalation to be established with greater confidence.

The animals studied here were somewhat younger and smaller than those who would normally be stunned using CO_2_. This was for practical reasons. We cannot say with certainty that younger pigs would respond similarly to those of slaughter weight, but we assume that the behaviours exhibited, and their order of appearance are the same. As we did not intend to determine the exact time point for the loss of consciousness, but rather the physiological changes present at the onset of various behaviours, we postulate that the problems with using younger pigs are minor.

## 5. Conclusions

We characterised 13 behaviours during CO_2_ administration to pigs. Four of these were exhibited by all the pigs in the current study. Vigorous behaviours were observed with physiological variables both compatible and incompatible with consciousness. Hence, we conclude that vigorous behaviours are not reliable indicators for loss of consciousness in pigs during CO_2_ stunning. This applies less to opisthotonos and agonal gasping as these behaviours occurred at a time where PaCO_2_ and pH in arterial blood in all pigs are within a range that is not compatible with consciousness. 

We also conclude that some of the adverse behaviours occurred at a time when some physiological variables were within a range compatible with consciousness. Given that some pigs showed vigorous muscular activity within the first 5 s of inhalation indicates that vigorous behaviours are unreliable indicators of unconsciousness.

We conclude that arterial blood pressure and heart rate remain within a range compatible with consciousness, even after the obvious loss of consciousness, thereby confirming that circulatory collapse was not the cause of the loss of consciousness.

The current study confirmed the findings of previous studies reporting rapid increases and decreases in PaCO_2_ and pH values, respectively, when pigs inhale high concentrations of CO_2_.

## Figures and Tables

**Figure 1 animals-13-02387-f001:**
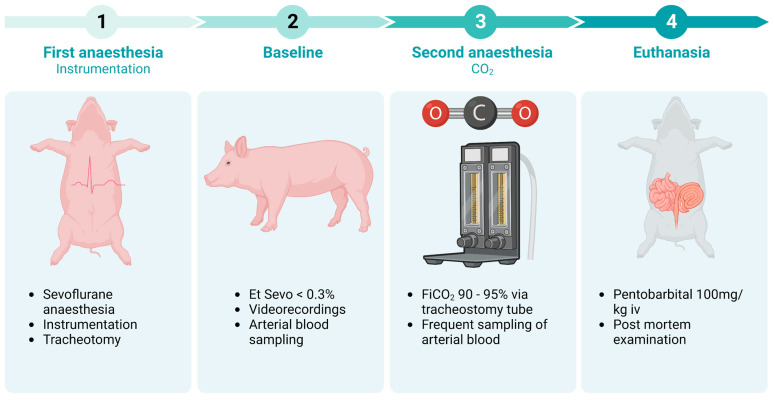
The figure illustrates the experimental timeline for the current study from first anaesthesia to postmortem examination. End tidal sevoflurane is abbreviated Et Sevo, and the fraction of inspiratory carbon dioxide is abbreviated FiCO_2_. Created with BioRender.com (accessed on 22 April 2023).

**Figure 2 animals-13-02387-f002:**
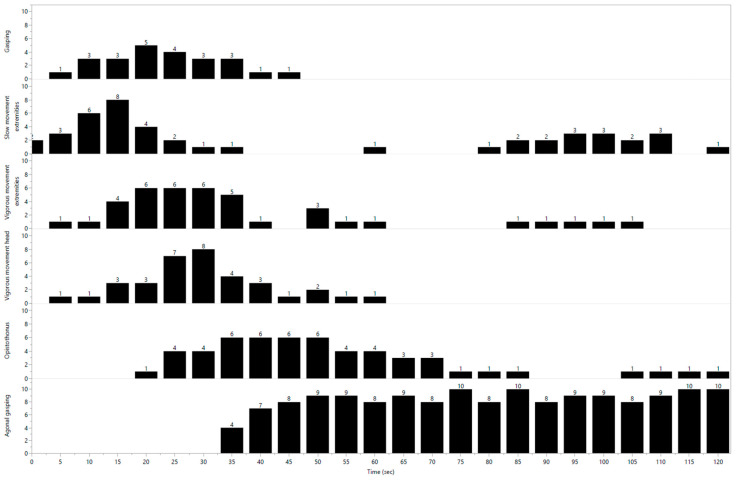
The *x* axis illustrates time from start of inhalation of 90–95% CO_2_. The number of pigs exhibiting each behaviour within a five-second period is given on the *y*-axis. Eleven pigs were studied. The behaviours are described in an ethogram (Table 1). The numbers on the top of the bars refer to the number of pigs exhibiting that specific behaviour within a given 5-s interval.

**Figure 3 animals-13-02387-f003:**
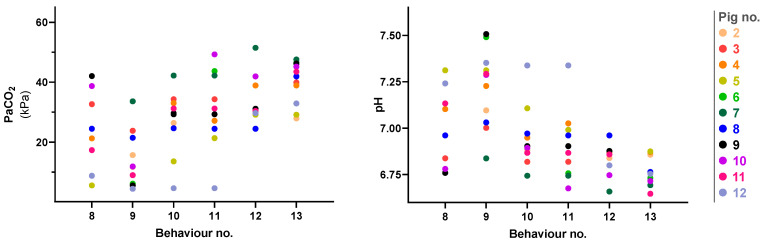
Figure illustrates PaCO_2_ (**left**) and pH (**right**) at the onset of key behaviours exhibited by pigs inhaling 90–95% CO_2_. Behaviours are numbered on the *x*-axis as follows: 8 = gasping, 9 = slow extremity movement, 10 = vigorous extremity movement, 11 = vigorous head movement, 12 = opisthotonos, 13 = agonal gasping. Each dot represents an individual pig, and each pig was assigned an individual colour, as shown on the right. The individual colours are consistently used for the pigs through all figures. Pig no. 2 accidentally received a fraction of inspired oxygen of 0.10. Additionally, the tracheostomy tube of pig no. 3 disconnected from the breathing system at 30 s and was reconnected at 40 s.

**Figure 4 animals-13-02387-f004:**
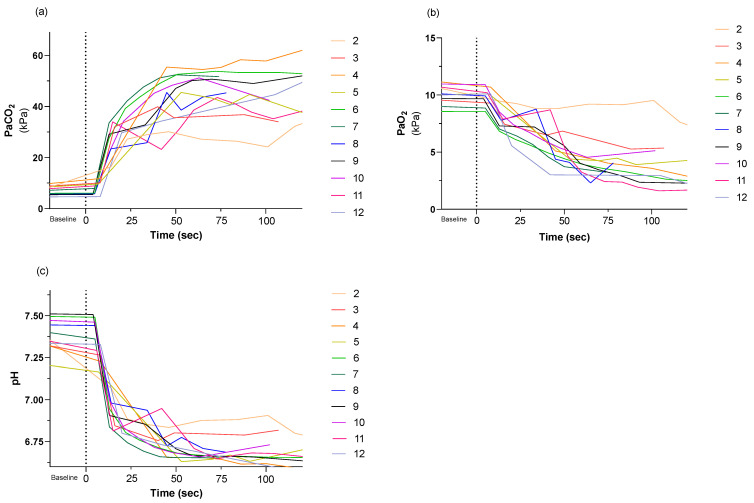
(**a**) Partial pressure of arterial carbon dioxide (PaCO_2_), (**b**) partial pressure of arterial oxygen (PaO_2_) and (**c**) pH of arterial blood from baseline and during CO_2_ inhalation. Administration of 90–95% CO_2_ began at time 0. Each line and number represent an individual pig, and each pig was assigned an individual colour, as shown on the right. The individual colours are consistently used through all figures. Pig 2 inspired a fraction of oxygen (FiO_2_: 0.10), and pig 3 accidentally became transiently disconnected from the breathing system.

**Figure 5 animals-13-02387-f005:**
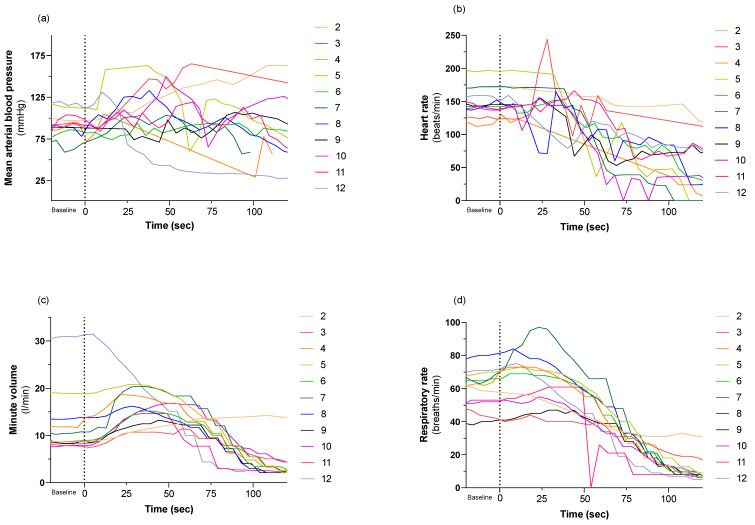
Cardiovascular and respiratory variables during baseline and CO_2_ inhalation. (**a**) Mean arterial blood pressure (mmHg), (**b**) heart rate (beats/min), (**c**) respiratory minute volume (litres) and (**d**) respiratory rate (breaths/min) are given on the *y*-axis and time (seconds) is given on the *x* axis. Administration of 90–95% CO_2_ started at time 0. Each line and number represent an individual pig, and each pig was assigned an individual colour, as shown on the right. The individual colours are consistently used through all figures. Pig 2 inspired a fraction of oxygen (FiO_2_: 0.10), and pig 3 accidentally became transiently disconnected from the breathing system.

**Table 1 animals-13-02387-t001:** Behaviours observed in video recordings of 11 pigs inhaling 90–95% CO_2_ after 5 min baseline recording. The characteristics and number of animals exhibiting each behaviour is given. Behaviours subjected to further analysis were given a shorter name, as stated in the last column.

Behaviour Number	Behaviour Characterized	Number of Animals Exhibiting Behaviour	Name
1	Lying calm in sling, head is in a relaxed posture	11	Calm
2	Lateral or dorsoventral movement of head	11	Slow head movement
3	Forceful expiration	3	
4	Large muscle groups are shivering	1	
5	Shaking head along the longitudinal axis	4	
6	Spontaneous chewing and/or tongue movements	4	
7	Spontaneous closing and opening eyelids	11	
8	Opening mouth wide while still breathing spontaneously	8	Gasping
9	Pendulating movements in extremities with a low frequency and no risk of displacement from the hammock	11	Slow extremity movement
10	Forceful galloping movements in extremities with a high frequency and amplitude. Risk for displacement from the hammock	10	Vigorous extremity movement
11	Forceful head movements with a high frequency and amplitude	10	Vigorous head movement
12	Continuous dorsoflexion of neck	9	Opisthotonos
13	Opening mouth wide and inhaling while having stopped breathing spontaneously in between mouth openings	11	Agonal gasping

**Table 2 animals-13-02387-t002:** The table gives the median (range) of some physiological variables at the onset of key behaviours in 11 pigs inhaling 90–95% CO_2_. The column marked *n* denotes the total number of pigs exhibiting the specific behaviour at some point during CO_2_ inhalation.

Behaviour	n	Time to Behaviour (s)	PaO_2_ kPa	K^+^ (mmol/L)	MAP (mmHg)	MV L/minute
Gasping	8	17 (0–44)	8.79 (6.14–9.87)	4.6 (4.3–6.4)	108 (71–131)	15.0 (9.3–29.9)
Slow extremity movement	10	8 (0–13)	9.68 (7.03–10.7)	4.4 (3.8–4.7)	90 (82–112)	10.3 (8.5–31.3)
Vigorous extremity movement	9	18 (5–26)	7.98 (7.17–9.97)	4.6 (3.9–5.0)	105 (88–158)	15.1 (10.1–31.4)
Vigorous head movement	10	23 (5–53)	8.15 (5.08–9.97)	4.6 (3.9–4.9)	112 (88–137)	16.4 (10.1–31.4)
Opistothonos	8	28 (20–43)	7.22 (4.38–8.82)	5.2 (4.0–5.8)	102 (87–162)	17.1 (11.5–26.0)
Agonal gasping	10	37 (31–73)	5.68 (2.43–8.65)	5.0 (4.5–7.7)	99 (56–162)	15.5 (7.7–20.7)

Time to behaviour: first appearance of given behaviour, PaO_2_: partial pressure of oxygen in arterial blood, K^+^: potassium concentration in arterial blood, MAP: mean arterial blood pressure, MV: respiratory minute volume.

## Data Availability

The data presented in this study are openly available in DataverseNO at https://doi.org/10.18710/R8XRZ4.

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
