# Peer review of "CO2 Stunning in Pigs: Physiological Deviations at Onset of Excitatory Behaviour"

_animals, 2023, doi:10.3390/ani13142387_

Round 1

Reviewer 1 Report

This is a prospective descriptive study in cross-bred pigs examining behaviours exhibited during inhalation of 90-95% CO2. Behaviors were noted, an ethogram developed, and physiological parameters were assessed during inhalation of CO2. While we cannot know what the individual pigs perceive during CO2 stunning, we must infer their experience from observations, prior publications, and extrapolations from the experiences of humans and other animals.

The study was not designed to determine the precise point of loss of consciousness with CO2 inhalation but rather to describe aversive behaviors observed during CO2 inhalation relative to PaCO2 and pH. While loss of consciousness is associated with increased spinal motor activity, the main findings are that early excitatory behaviour is not a reliable indicator for loss of consciousness in pigs during CO2 stunning and that PaCO2 increases rapidly and pH declines rapidly to levels associated with unconsciousness within 20-25 seconds and the observation of opisthotonos and agonal breathing.

Unlike usual whole body immersion methods employed for these kinds of studies, the authors chose to first place a tracheostomy tube and IV and IA catheters under sevoflurane anesthesia and, after a 30 min sevoflurane washout period, expose the pigs to a high concentration of CO2 through the pre-placed tracheostomy tube. This effectively eliminates the sharp acidic smell of CO2 and any potential interaction with ocular and mucous membranes from the observed behaviors.

Specific comments follow.

Line 53: “Previous studies have showed that exposure to high concentrations of CO2…” Reference 3, Raj and Gregory 1995, demonstrated that pigs found exposure to 30% CO2 no more aversive than air in their cross over study.

Line 115: A minor point, but the O2 can’t be 100% if sevoflurane is added. Delete “100%”

Line 148: “was” is misspelled

Line 281-282: “For the behaviours gasping…” Please provide the PaCO2 and pH values here. It was awkward trying to extrapolate them from Figures 3 and 4. Would it be possible to provide a table with PaCO2 and pH similar to Table 2?

I can accept that Pig 12 was likely conscious with a PaCO2 of 4.61 kPa but the water is a bit muddier with Pig 5 and a PaCO2 of 13.54 kPa. Consider the following:

Eisele et al. (ref 25) states that PaCO2 >95 mmHg (13 kPa) and a pH <7.10 is progressively narcotic and that 30% CO2 is anesthetic (228 mmHg; 30 kPa.

Ueda cites Eisele that 30% CO2 is anesthetic (Ueda I. Anesthesia: an interfacial phenomenon. Colloids and surfaces. 1989 Jan 1;38(1):37-48.)

Ramirez claims CO2 is as potent an anesthetic as N2O according to the Meyer-Overton lipid solubility theory (Ramirez RM. A complex life habitable zone based on lipid solubility theory. Scientific Reports. 2020 May 4;10(1):1-8.)

Scott et al. reviewed acute CO2 exposure in humans. They reported that at levels of 17% and above, CO2 causes convulsions, unconsciousness, and death in seconds. In studies reported by the National Institute for Occupational Safety and Health (NIOSH) and reviewed by Langford (Langford NJ. Carbon dioxide poisoning. Toxicological reviews. 2005 Dec;24:229-35), unconsciousness was experienced at levels of 10% and above for a duration of 10–20 min. Eye flickering, myoclonic twitches, dilated pupils and restlessness were reported for exposures of 10–15% for periods of 90 seconds, and the hearing threshold was reduced at levels of 38%. Narcosis was reported to occur in a matter of seconds when carbon dioxide is present at levels of 30% (Scott JL, Kraemer DG, Keller RJ. Occupational hazards of carbon dioxide exposure. Journal of Chemical Health & Safety. 2009 Mar 1;16(2):18-22.)

Scott et al. 2009 also cited a report of accidental human CO2 exposure. The individual who was overcome later said that when he entered the contaminated area he was aware that breathing was difficult, but had no warning signs before becoming unconscious. A senior chemist conducted air sampling about an hour and a half after the incident occurred and found elevated levels of CO2 present in the ship’s hold. Later experiments showed that the atmosphere where the incident occurred probably contained at least 25–30% CO2.

In light of these publications, an alternative interpretation is that some pigs may be in the initial stages of anesthesia during early onset excitatory behaviors when exposed to stunning levels of CO2. 

Line 319-322: This gets into the use of MAC reduction as a proxy for unconsciousness. To support your claim, Aranake et al., 2013 states (p. 519) that MAC is not a reliable indicator of hypnosis or unconsciousness. However, because loss of motor responsiveness occurs at a higher dose than unconsciousness, MAC is a very good pragmatic end point as it gives the anesthetist reasonable confidence that the patient is likely to be unconscious.

Reviewer 2 Report

The purpose of this paper seems to be to contribute to the consideration of the welfare consequences of CO2 stunning. As such I feel that some discussion should be devoted to:

·       The implications of using young pigs rather than slaughter weight pigs,

·       the implications of using a tracheotomy thereby circumventing the pain and distress caused by carbon dioxide in the nasal cavities, and

·       the implications of the almost instantaneous introduction of very high concentration CO2 rather than the more gradual introduction experianced in a pit.

A key question that this paper apears to aim to  answer is whether observed behaviours  occur before the loss of consciousness and so may be indications of distress, or occur after the loss of consciousness. However despite the depth and care with which the behaviours are observed, the point at which consciousness is considered to lost is given almost no consideration. We see that the authors are linking it to blood pH and  CO2 content however I could find no discussion of what levels they are considering to be compatible with consciousness, nor any reference to where these levels might have come from. This appears to be a serious weakness on this report. I agree with the authors that we have only a poor idea of when consciousness is lost however if statements such as “ Our study objective was to investigate loss of consciousness by linking physiological variables to 28 the start of behaviours observed during CO2 inhalation.” Then we need to know how they are determining this point.

Another area I would like the authors to give consideration to is the definition of their behaviours. There is acceptance in the paper that behaviours described by other authors are unclear (eg L 292), but we find the same problem in this paper. What exactly is “vigorous movement” and slow movement. How large and how fast are these.?  

Towards the end of this paper the authors create a category of behaviours which they call “Excitory behaviours” Which ae these?

I would ask the authors to give consideration to  combining Table2 and Figure2 to put cause and effect onto the same figure.

This paper is generally well presented with a good standard of english. There are one or two typos and/or grammatical errors that will be captured by a standard grammar and spell checkerand a few poorly chosen words such as  Mandatory (l 292) and  Ensued (l 413) which would be captured on a read through by a native English speaker – or otherwise can be ignored.

Reviewer 3 Report

How well does a 30 kg pig represent a market weight pig? Otherwise sound science.

English language problems are highlighted in the attached file.

The study contributes to the understanding of pig welfare under carbon dioxide stunning. This is a major concern since this stunning method is becoming common. In considering welfare, the range of responses is as or more important than the mean. In particular, the study addresses the relationship of observed behaviours to the state of consciousness, and their relative timing.   The methods are appropriate and clearly described, the results are well presented, and the conclusions are supported by the results.   Overall, the study makes an important contribution to the assessment of consciousness and welfare under carbon dioxide stunning.

English language problems include subject-verb agreement and incorrect word use (eg in conjunction with)

Author Response

Please see attechment

Reviewer 4 Report

Hognestad et al. CO2 stunning in pigs: Physiological deviations at onset of excitatory behaviour.

General comments: The purpose of this study was to describe several physiological variables at the onset of pertinent behaviours indicative of distress observed in instrumented pigs restrained in a sling during CO2 inhalation.

Specific comments:

The text does not flow well and there are moderate grammar errors throughout.

Throughout the paper the authors have used the terms ‘conscious’ and ‘unconscious’; however, no EEGs were collected. Even with EEGs it is difficult to determine when consciousness is present in some human patients, let alone animals. For this reason, this reviewer prefers the terms ‘sensibility’ and insensibility’, as these are more accurate descriptors of what is being tested and signalled – that is, responsiveness to stimulation. I would encourage the authors to consider using these terms to enhance the accuracy and clarity of their work.

Methods:

Pg 3, line 95 – please include how the sample size was determined for the experiment.

Pg 3, line 133 – please include information on the average duration of anesthesia for induction and instrumentation. It is difficult to know whether a 30min recovery period post-surgery was sufficient without this information.

Pg 4, line 148 – were tie ropes used to restrain the pigs in the sling? Please describe how the sling was sourced, its size, and composition.

Pg 4, line 159 – please describe how the flow rate for CO2 delivery to be used was determined. The rates seem very high and exceed those recommended in the AVMA Guidelines on Euthanasia.

Pg 4, line 176 – please provide more detail about viewing/scoring the videos – how long were the recordings, was the observer blinded to video number and timing of gas or was this obvious. Were videos of baseline and gassing randomized, etc? Please describe the cameras in more detail in the paragraph above.

Results:

Overall, the text is overly descriptive and no attempt has been made to summarize key findings or to present mean results or trends. Data transformation may be needed for this to better describe and characterize mean relative changes from baseline. Again, it is hard for the reader to gauge what was going on with the animals from the minimal descriptive information given. The results section should fully describe the findings of the study and put them into context against the research question(s).

Pg 10, line 267 – it is noted that 7 of 11 pigs underwent histopathology evaluation but this is not described in the Methods. Please describe more fully in the methods the pathology – who conducted the post mortems, how tissues were processed and stained and who read the slides. How were animals randomly selected for histopathology and how long after death did post mortem evaluation and sample collection occur? The authors note on pg 12, line 373 of the Discussion that pathology findings are beyond the scope of the article, but they can’t be only partially included and partially described and interpreted.

Pg 10, line 281 – please compare the findings with published literature that corroborate pH and PaCO2 levels compatible with sensibility. Is there room for doubt? Is there other literature that suggests they might also be insensible?

In tables and text, please note that ‘nr’ is a nonstandard and informal abbreviation of the word ‘number’. It should be replaced with ‘no.’ throughout.

Pg 10, line 263 – please indicate that these observations were made at necropsy.

Discussion:

The purpose of the discussion is to interpret and describe the significance of the findings in light of what was already known about the research problem being investigated, and to explain any new understanding or fresh insights about the problem after you've taken the findings into consideration. How has this study moved our understanding of pig response to CO2 inhalation forward?

Pg 10, line 271 & following – results are presented for the first time about physical signs. These should be moved from the Discussion and placed at the appropriate point in the Results section.

Pg 11, paragraphs 2 & 3 provide a summary of results for Figures 4 and 5 that should instead be included in the Results. The following text explains findings from individual pigs, instead of addressing the overall summary and relevance of the group findings.

Pg 12, line 377 – the authors justify their approach in describing individual pig reactions as being inclusive of a range in responses for a population, and suggest that using more animals would be ethically irresponsible and too expensive. This is nonsense, especially with a small sample size and no justification for how group sizes were determined. What was the value in conducting this project if no meaningful data can be derived that can be applied to a population?

In the figure and Table headers please include the number of pigs in the study.  Please add arrows to Figure 2 to indicate when the gas is turned on.

Table 1 – the use of nonstandard descriptors in this ethogram makes it very difficult to compare the results of this study with that of others.

Table 2 – does K represent potassium ion? The table footnote indicates Kalium (not an English word).

Figure 3 – x axis label – Correct ‘nr’. This image is very difficult to interpret because it leaves out the dimension of time. This reviewer doesn’t see any need to include this as all information is already present within Table 2.

Figure 4 – please label the 3 panels as a, b, and c. Please add descriptive info for Figure 4 that includes mean values for PaCO2, PaO2, and pH, and discuss the changes observed over time in the text. The same comments apply to Figure 5 – what are the overall results and what is the reader supposed to ‘take home’ from these panels (please label the panels to better direct the reader when describing the results).

It is unclear whether there are Supplementary materials – I couldn’t find any to download but the statement suggests that there are some. The data availability statement has not been completed.

Please see above.

Round 2

Reviewer 4 Report

Thank-you for partially addressing reviewer comments. This reviewer still has concerns regarding study design, including calculation of sample size, and the lack of even rudimentary descriptive statistics to help interpret findings.

Awkward phrasing remains in sections. Unclear what this sentence means: "Prior to inhalation of carbon dioxide nine pigs exhibited behaviour one, calm, ...." What is 'behaviour one'?

Author Response

Dear reviewer,

Thank you again for your comments. We are pleased to inform that our manuscript now have gone through mdpi english editing service, and we hope that the language now is meets your expectations. 

Regarding your comment on sample size we have provided an explanation for this in the manuscript, and regarding statistics we have detailed several places in the manuscript why we have not performed statistics other than median and range as we want to adress the individuals. We acknowledge your point of view, but we still believe this is the right way to present these results.